# Contact Mixing Patterns and Population Movement among Migrant Workers in an Urban Setting in Thailand

**DOI:** 10.3390/ijerph17072237

**Published:** 2020-03-26

**Authors:** Wiriya Mahikul, Somkid Kripattanapong, Piya Hanvoravongchai, Aronrag Meeyai, Sopon Iamsirithaworn, Prasert Auewarakul, Wirichada Pan-ngum

**Affiliations:** 1Department of Fundamentals of Public Health, Faculty of Public Health, Burapha University, Chon Buri 20131, Thailand; wiriya.ma@buu.ac.th; 2Bureau of Epidemiology Department of Disease Control, Bangkok 11000, Thailand; skongyu@gmail.com; 3Department of Preventive and Social Medicine, Faculty of Medicine, Chulalongkorn University, Bangkok 10330, Thailand; piya.h@chula.ac.th; 4Department of Epidemiology, Faculty of Public Health, Mahidol University, Bangkok 10400, Thailand; a.meeyai@gmail.com; 5Department of Global Health and Development, London School of Hygiene & Tropical Medicine, London WC1E 7HT, UK; 6Department of Disease Control, Ministry of Public Health, Bangkok 11000, Thailand; sopon@ddc.mail.go.th; 7Institute of Molecular Biosciences (MB), Mahidol University, Nakhon Pathom 73170, Thailand; prasert.aue@mahidol.ac.th; 8Department of Tropical Hygiene, Faculty of Tropical Medicine, Mahidol University Bangkok, Bangkok 10400, Thailand; 9Mahidol-Oxford Tropical Medicine Research Unit, Faculty of Tropical Medicine, Mahidol University, Bangkok 10400, Thailand

**Keywords:** contact mixing patterns, migrant workers, urban setting

## Abstract

Data relating to contact mixing patterns among humans are essential for the accurate modeling of infectious disease transmission dynamics. Here, we describe contact mixing patterns among migrant workers in urban settings in Thailand, based on a survey of 369 migrant workers of three nationalities. Respondents recorded their demographic data, including age, sex, nationality, workplace, income, and education. Each respondent chose a single day to record their contacts; this resulted in a total of more than 8300 contacts. The characteristics of contacts were recorded, including their age, sex, nationality, location of contact, and occurrence of physical contact. More than 75% of all contacts occurred among migrants aged 15 to 39 years. The contacts were highly clustered in this age group among migrant workers of all three nationalities. There were far fewer contacts between migrant workers with younger and older age groups. The pattern varied slightly among different nationalities, which was mostly dependent upon the types of jobs taken. Half of migrant workers always returned to their home country at most once a year and on a seasonal basis. The present study has helped us gain a better understanding of contact mixing patterns among migrant workers in urban settings. This information is useful both when simulating disease epidemics and for guiding optimal disease control strategies among this vulnerable section of the population.

## 1. Introduction

Epidemiological models of infectious diseases have tended to assume a simple structure for population mixing patterns because data on this subject are generally not widely available [1,2,3]. Several non-random mixing patterns, such as proportional and non-proportional mixing, have been incorporated into mathematical models to represent sufficient heterogeneous contacts among a population [1,4]. The objectives of mathematical models can range from simply studying how a disease spreads to the design of disease control strategies. The development of good models requires validation and sensitivity analyses. Sensitivity analyses have often suggested that model outputs, for example spreading patterns or total numbers of cases, were sensitive to uncertainties in model parameters such as the contact matrix, a qualitative measure of the contact pattern [5].

More studies have focused on sexual mixing patterns [6,7,8,9], but only a few have looked at close contact patterns, which are an important element of airborne infections [10,11]. Contact patterns among the Thai population of all ages, living in four different regions (central, north, northeast, and south) were investigated during a study conducted in 2009 [12]. The contact frequency was highest among children aged 5 to 15 years. Almost 80% of contacts were physical in the younger age groups, but this figure decreased to less than 40% in teenagers and adults. Approximately 50% of all contacts occurred at home, and this percentage was higher among infants and elderly people. The results showed a similar pattern to those seen in previous studies from many countries, i.e., that human contact patterns are highly age-dependent, with greater frequencies and some slight variations among the different regions of the country, which could in part be explained by cultural differences [10,13,14,15,16].

A previous social contact network study between Thailand and the Netherlands showed that mixing patterns were assortative by demographic variables such as age, sex, and education level, and random by total number of contacts [17]. Most workers from Myanmar had never returned to their home country, but Cambodian workers returned home frequently [18]. Key migrant workers in Thailand are Myanmar, Cambodian, and Laos PDR. Migration has been linked to disease transmission patterns. Studies related to migrants is challenging for various reasons, including international politics and legal entry and work permit issues. Yet it is essential to characterize their contact mixing patterns to learn about infectious disease epidemiology and plan optimal control strategies for this group of population when there is an outbreak. The study aimed to describe some important characteristics of migrant workers in the urban setting of Thailand including their contact mixing patterns and movement. This study focused on the migrant workers living and working around the suburb areas of Bangkok. This population is an important labor force to industry, especially under the ASEAN Economic Community (AEC) where free movement of goods, services, and skilled labor has been established. A better understanding of the contact mixing patterns and population movement among this population will guide how to control infectious disease spreading in an outbreak and how to design health strategies to prevent future infections. 

## 2. Materials and Methods 

### 2.1. Study Area and Data Collection

The survey of social contacts and mixing patterns was performed using a questionnaire-based interview of participants to collect their demographic data. Only participants who gave written, informed consent were recruited to the study. The interviews were conducted by trained research staff and translators using a questionnaire translated from English. The translators explained to the participants in this contact mixing study how to keep a diary and how to record their contacts. The research team then arranged to collect the diary cards the next day after the participants made a 24 h record of their contacts, and finally the contact mixing patterns were extracted from the survey data.

The survey was conducted in various workplaces in Pathum Thani, Thailand, between September and November 2015. The target population was migrant workers from Cambodia, Lao People’s Democratic Republic (Lao PDR), and Myanmar who were willing to participate and signed informed consent forms. On return of the completed diary card, 10% of participants were chosen at random for interview, to fill in any missing contact details. A contact was defined as either skin-to-skin contact or a two-way conversation between individuals who were approximately one meter apart and without skin-to-skin contact. For each contact, the participant would be asked to record information about their age group, sex, nationality (same or different from the participant), type of contact (physical or non-physical), and the location where the contact occurred. The diary design was modified from that used in the POLYMOD study, which characterized contact patterns in Europe [10]. The information asked for was simplified and minimized the use of written answers because some migrants were illiterate. The questionnaire and diary card were in English originally. It was professionally translated from English into three different languages (Myanmar, Cambodian, and Lao) by the translators who had experience working in the health-related sector. The translated version was then validated independently by the second person (the international students at Mahidol university) for correctness and appropriateness of the terms used.

Pathum Thani is a province that borders the capital of Thailand, Bangkok (Figure 1); it has an area of 1525 km^2^ and a registered population of more than one million people. The province includes a mixture of agricultural and industrial areas; the latter reflects the rapid growth of the manufacturing industry in Thailand. Pathum Thani is representative of urban settings where migrant workers live and work in large cities around Thailand. Other settings where there are large numbers of migrant workers are the cross-border areas, which were not the focus of this study. The number of registered migrants in Pathum Thani in 2015 was 138,373, which comprised approximately 9.5% of all migrant workers in Thailand that year. The ratio of male to female migrant workers was 1.67:1. By nationality, 60% were from Myanmar, 28% were from Cambodia, and 12% were from Lao PDR [19]. There is also a significant number of unregistered migrant workers working in Thailand and Pathum Thani, whose actual number is unknown. It is estimated that there were approximately 600,000 unregistered versus 1,200,000 registered migrant workers, a ratio of about 2 to 1 [20].

Purposive sampling was applied to achieve a representative selection of migrant workers in an urban setting in Thailand; we focused on those who worked in industrial factories, wholesale vegetable markets, other kinds of factories, and finally those who worked at home. The four districts in Pathum Thani (Klong Luang, Lum Look Ka, Lad Loom Kaew, and Klong Sam) comprised the study sites. Sampling was performed in such a way so as to ensure that migrant workers from all three sizes of factories were included, i.e., small factories (up to 20 employees), medium factories (between 21 and 50 employees), and large factories (more than 50 employees).

The inclusion criteria were as follows: 1) Originally from Cambodia, Lao PDR, or Myanmar; 2) had lived in Thailand for more than 3 months; 3) aged more than 15 years old; and 4) gave written, informed consent. The exclusion criteria were as follow: 1) Migrants who were not willing to answer questions; 2) women who were pregnant; and 3) individuals with underlying diseases, including immunosuppression, kidney disease, liver disease, and cancer, whose patterns of contact mixing may not be good representatives of the population.

This study was approved by the Ethical Review Committee for Human Research, Faculty of Public Health, Mahidol University, MUPH 2015–117. Written informed consent was obtained from all of the participants in the study.

### 2.2. Data Analysis

Statistical analysis was performed using Stata/SE 14.0 for Mac and R software version 3.2.3. Descriptive statistics were presented as the absolute frequency (number) and relative frequency (percentage) for categorical variables and as mean and standard deviation (SD) for continuous variables. Negative binomial regression was used for assessing the main effects of covariates, including age, sex, nationality, workplace, income, and education on the number of reported contacts [21]. The relative number of reported contacts and 95% confident intervals are presented for all factors.

## 3. Results

The total sample size was 369 participants, which represents a response rate of 90%. Of these, 160 respondents were from Myanmar, 145 respondents were from Cambodia, and 101 respondents were from Laos PDR. The demographic characteristics of all participants and nationalities are shown in Table 1. The majority of migrant laborers were aged between 20 and 40 years old, with males comprising 48.5%. The mean age among migrant workers was 28.5 (25–35) years old. The smallest age distribution was seen among Cambodians, who were aged between 18 and 49 years, because they generally worked in construction or factories. Their age and sex distribution were significantly different (*p* < 0.01). There were significantly more females than males among migrant workers from Laos. Most of the migrant workers had graduated from primary school and received a daily wage of about 320 Thai baht. Most Cambodian migrant workers worked in food factories, the majority from Laos worked in markets, while those from Myanmar worked in various sectors, e.g., construction sites and waste-sorting sites.

Most of the migrant workers spent most of their time working as they got pay daily (only 54% of them had one day off each week). Workers from Myanmar and Laos were engaged in the largest variety of work. Fifty percent of respondents reported that they returned to their own country at most once a year on a seasonal basis, e.g., during the local or the Western new year period; workers from Myanmar returned to their home country less often than those from other countries (Table 1).

A total of 8356 contact persons were reported (Table 2). The highest number of contacts occurred with migrant workers from Myanmar, with 37% of all contacts. However, the average number of contacts per person for Myanmar and Laos was similar, i.e., around 15 times per day and relatively lower for Cambodia, i.e., 12 times per day. A large number of contatcs was with people aged 14 to 40 years (75%), followed by those aged more than 40 years (16%), 5 to 15 years (6%), and less than 5 years (3%). The proportion of physical contacts varied from 18% to 38%. Migrant workers from Myanmar had the most physical contacts among all nationalities, while those from Laos had the least. For all nationalities, more than half of all contacts occurred in the workplace. Contact mixing patterns across age groups are shown in the contour plots (see Figure 2). Contacts from work resulted in the highest density among migrants aged 15 to 39 years. The patterns of contacts among migrant workers from Myanmar and Cambodia were more similar to each other than those of Laos migrants (see Figure 3). Migrant workers from Laos worked more from home or markets, which made it easier for them to go to other places.

According to the contact mixing patterns by age and type of work (see Figure 4), there was a significantly different pattern based on their type of work. Migrants who worked in the markets and other kinds of factories had frequent contacts with others aged between 15 and 39 years old. The contact mixing pattern of migrant workers who worked in food factories (Figure 4A) and those employed to do housework (Figure 4B) were completely different from the common pattern where contacts were highly assortative with age.

Analysis of the total number of reported contacts using a multiple negative binomial regression model showed that Cambodian migrant workers had a significantly lower number of reported contacts, about 16%–22% compared with Myanmar migrant workers, as shown in Table 3. Agricultural workers had a significantly higher number of reported contacts, with 46%–67% more contacts than other occupations. We observed no associations between the total number of recorded contacts and gender, age, type of work, education level, or income per day.

## 4. Discussion

Infectious disease prevention and control policies are increasingly being evaluated using mathematical models. For instance, household size, class size, transport statistics, and workplace size distribution are all parameters that have been used in recent mathematical models to define contact pattern structures [22,23,24,25]. Contact mixing patterns among the Thai population of all ages, living in four different regions (central, north, northeast, and south), were investigated during a previous study conducted in 2009 [12]. That study used cross-sectional surveys conducted by different commercial companies or public health institutes. Our study adopted a similar method but with different target populations and workplaces, i.e., migrants in different workplaces. The previous also showed that contact mixing patterns were highly assortative by age: Schoolchildren and young adults in particular tended to mix with people of the same age. 

A recent study by Ajelli and colleagues [26] also showed that contacts were highly assortative by age, especially for school-age individuals, and that the number of contacts was negatively correlated with the age of participants. Kiesha and colleagues [15] showed that contacts are highly assortative by age within a household, while Asian people had more inter-generational contacts than in other nationalities. In this study, we found that more than three-quarters of all contacts were highly clustered among migrants in the 15 to 39 years old age group. One important finding from our study is that the age and intensity patterns of contacts are remarkably similar among migrants, even though the average number of contacts differed. Another major insight gained from our study comes from the observation that the contacts made among those of working age are more assortative than contacts made by other age groups. Leung and colleagues showed that there was a high level of contact among the population aged 25 to 60 years, for both participants and contacts, involving contacts at the workplace [27], while our findings showed a similar pattern, in that more than 50% of all contacts occurred in the workplace among all nationalities. 

A previous study in a European setting showed that a greater frequency of contacts and some slight variations among the regions of a country could be partly explained by cultural differences [10], which is similar to our finding that Cambodian migrant workers were associated with a significantly lower number of reported contacts compared with the number reported by Myanmar migrant workers. In addition, people from Laos had more diverse contacts, with different age groups and nationalities. This can be mainly explained by the type of work they were involved in, and because people from Laos had the least language and cultural barriers when mixing with the general Thai population. Migrant workers from Laos often take up work where communication and human interaction skills are more sought after, such as shop assistants, fruit and vegetable vendors, housekeepers, and children’s nannies. Conversely, workers from Cambodia often operate machinery in factories and in industry or are employed as construction workers. It is also possible for them to work in markets, often in jobs where no interactions with people are required, such as preparing vegetables. Melegaro and colleagues showed that the number of social contacts among the general population was higher in farming areas compared with the contacts in peri-urban townships [28], which is similar to our finding that agricultural workers had significantly more contacts than employees. 

Fifty percent of migrant workers returned to their home country at least once a year and on a seasonal basis, for example during the Thai or the Western new year period, which is similar to findings from a previous report [18,29]. Migrant workers from Myanmar returned to their home country less often that those from other countries.

Our study has some limitations. Research in migrant workers is generally very challenging and required cooperation from the employers. Purposive sampling was selected to achieve a representative selection of migrant workers in a Thai setting; however, this may contribute to the limited information on the contact mixing of general migrant workers. Using contact diaries completed by individuals in the general population was a feasible method for our specific study objectives. However, as with all self-reported data, future research should be conducted that uses different approaches, including interviews or direct observation, to validate our findings. Alternatively, the use of synthetic generation of age-stratified mixing pattern may be explored when the direct measure of contacts may be difficult [30]. This study has some gaps, i.e., because we focused on studying the contact mixing patterns among the migrant workers, we missed the age groups that represent children and elderly in this study. In addition, we did not have information on formal registration against unregistered migrants to analyze the differences across these groups. 

## 5. Conclusions

There were significant differences among the contact mixing patterns of migrant workers in an urban setting in Thailand. These differences were not so apparent by nationality, but these differences could be clearly seen on the basis of the type of work migrant workers were engaged in. Migrant workers from Cambodia and Myanmar had more similar contact mixing patterns to each other compared with the contact mixing patterns among workers from Laos. Contact was heavy among the age class 15–39 of all three nationalities. The number of social contacts among the agricultural workers was higher than among the employees. Half of those workers always returned to their own country at most once a year and on a seasonal basis. The findings of this study can help us and public health practitioners gain a better understanding of the contact mixing patterns and population movement among migrant workers in an urban setting. This information will be useful for simulations of disease epidemics. The findings will also help to improve the parameterization of mathematical models for infectious disease transmission in this specific population and guide optimal disease control strategies among this sensitive and vulnerable section of the population.

## Figures and Tables

**Figure 1 ijerph-17-02237-f001:**
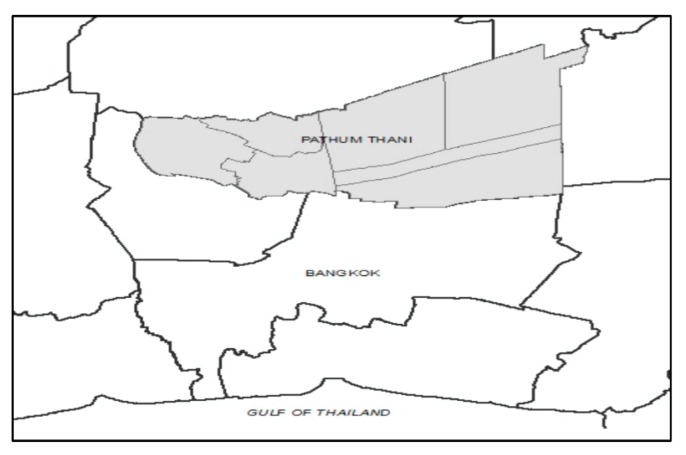
Pathum Thani, the study site.

**Figure 2 ijerph-17-02237-f002:**
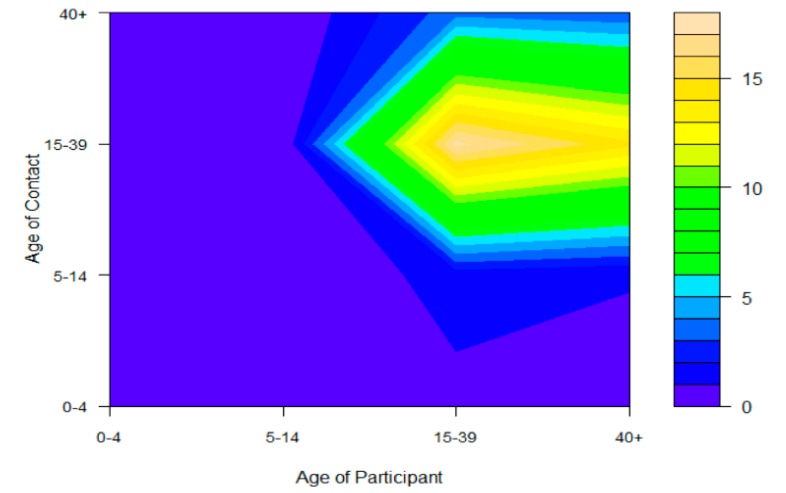
Overall contour plot of contact mixing patterns by age; yellow indicates a high number of contacts, green an intermediate number of contacts, and blue a low number of contacts.

**Figure 3 ijerph-17-02237-f003:**
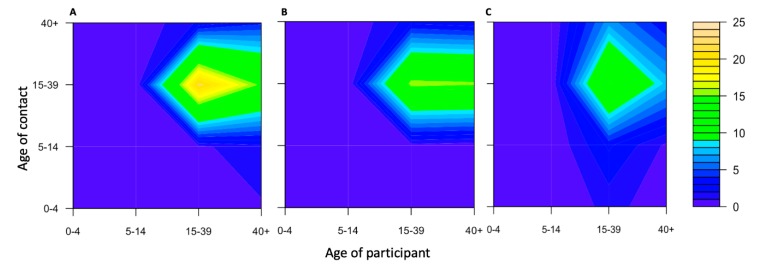
Contour plots of contact mixing patterns by age and nationality of the participants: (**A**) Myanmar, (**B**) Cambodia, and (**C**) Lao PDR; yellow indicates a high number of contacts, green an intermediate number of contacts, and blue a low number of contacts.

**Figure 4 ijerph-17-02237-f004:**
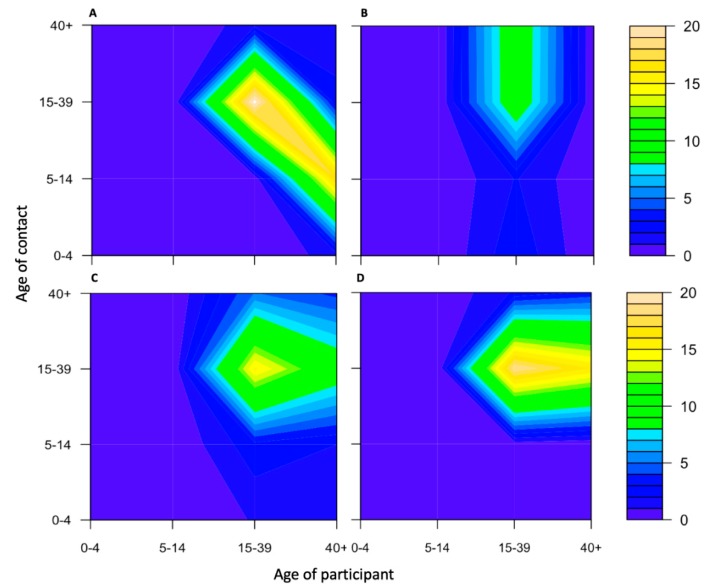
Contour plots of contact mixing patterns by age and type of work. (**A**) Food factory. (**B**) Housework. (**C**) Market. (**D**) Other kinds of factories; yellow indicates a high number of contacts, green an intermediate number of contacts, and blue a low number of contacts.

**Table 1 ijerph-17-02237-t001:** General information about participants by nationality.

	Myanmar	Cambodia	Laos PDR	Overall
**Demographics**				
Total number of participants	128	135	106	369
Mean age (SD)	28.95 (7.90)	29.34 (6.22)	26.95 (7.16)	28.52 (7.16)
Minimum age	15	18	16	15
Maximum age	52	49	59	59
Male (%)	63.28	54.07	23.58	48.51
Primary school only (%)	33.59	46.67	54.72	50.14
Daily wage in THB (SD)	315.87 (54.98)	304.73 (27.81)	360.57 (76.68)	323.02 (58.19)
General labor (%)	87.50	90.37	77.36	91.33
**Type of work**				
Food factory (%)	23.44	49.67	0	25.20
Private home (%)	0	0	16.03	4.61
Market (%)	25.00	22.96	83.94	41.19
Other kinds of factories (%)	51.56	30.37	0	29.00
**Population movement**				
One day off per week (%)	72.66	64.44	19.81	54.47
Return home every year (%)	25.78	77.78	44.34	50.14

**Table 2 ijerph-17-02237-t002:** Contact information reported on the diary cards, by nationality.

	Myanmar	Cambodia	Laos PDR	Overall
Total contacts	3128	2574	2654	8356
Contact with males (%)	64.87	55.83	45.48	55.92
Physical contacts (%)	37.79	26.92	17.75	28.07
Mean number of contacts/person (SD)	16.19 (11.61)	12.62 (9.40)	15.68 (11.35)	14.93 (11.00)
Number of contacts aged less than 5 years	59	53	137	249
Number of contacts aged 5 to 14 years	117	81	326	542
Number of contacts aged 15 to 40 years	2469	2109	1518	6276
Number of contacts aged more than 40 years	303	331	673	1307
Contacts at home (%)	42.39	40.95	28.15	37.42
Contacts at work (%)	67.30	56.06	68.61	64.25
Contacts in other places (%)	8.31	7.78	12.17	9.37

**Table 3 ijerph-17-02237-t003:** The relative number of contacts based on the multiple negative binomial regression model.

Category	Number of Participants	Mean (Standard Deviation) Number of Reported Contacts	Relative Number of Reported Contacts (95% Confidence Intervals)
**Gender**			
Male	179	22.37 (13.02)	1
Female	190	22.89 (10.70)	1.01 (0.89–1.13)
**Age**			
0–20	54	24 (13.29)	1
21–40	292	22.69 (11.67)	0.99 (0.86–1.16)
>40	23	18.78 (10.55)	0.82 (0.64–1.06)
**Nationality**			
Myanmar	128	24.43 (13.07)	1
Cambodia	135	19.06 (9.96)	0.76 (0.66–0.87) *
Laos	106	25.03 (11.60)	1.09 (0.90–1.31)
**Type of work**			
Food factory	93	22.39 (12.38)	1
Market	152	23.55 (11.82)	0.92 (0.77–1.10)
Private home and Others	124	21.71 (11.55)	0.98 (0.84–1.14)
**Education**			
Primary school	164	22.39 (11.86)	1
High school	155	23.49 (11.91)	1.08 (0.96–1.21)
Undergraduate	27	20.77 (12.78)	0.92 (0.74–1.15)
Postgraduate	2	17.5 (0.70)	0.89 (0.43–1.84)
Unknown	21	21.23 (11.21)	0.58 (0.34–1.01)
**Occupation**			
General labor	316	22.06 (11.64)	1
Merchant	18	24.61 (10.80)	1.23 (0.95–1.58)
Agricultural worker	11	36.90 (12.89)	1.59 (1.17–2.16) *
Other	3	29.66 (13.86)	1.24 (0.70–2.21)
Unknown	21	21.23 (11.21)	0.58 (0.34–1.01)
**Income per day**			
Less than 300 baht	239	21.94 (11.71)	1
300 baht or more	104	24.39 (12.35)	1.07 (0.94–1.23)
Unknown	26	22.07 (11.11)	1.47 (0.93–2.32)

* *p*-value < 0.05.

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
