# Peer review of "Contact Mixing Patterns and Population Movement among Migrant Workers in an Urban Setting in Thailand"

_ijerph, 2020, doi:10.3390/ijerph17072237_

Round 1

Reviewer 1 Report

The authors may try the statistical models developed in this paper https://arxiv.org/abs/2003.01214, which may help better understand the contact matrix obtained from the survey

Reviewer 2 Report

Contact mixing patterns among humans are essential for the accurate modelling of infectious disease transmission dynamics. Further, migration has been linked to the disease transmission patterns. It is important to characterize the migrants contact mixing patterns to prevent and control the infectious disease occur and outbreak. This study describes contact mixing patterns among migrant workers in urban settings in Thailand, based on a survey of 369 migrant workers of three nationalities. The paper is an interesting study. My major concerns about the study are:
1. The study has a weakness study design. There was about 138,373 registered in Pathum Thani in 2015. The study used Purposive sampling of total 369 subjects to achieve a representative selection of migrant workers in an urban setting in Thailand. The results contributed limited information to the contact mixing in the migrant worker populations.

  1. The definition of contact in the study was based on what?
  2. How to keep the quality in the questionnaire investigation and data analysis?
  3. According to the inclusion criteria, the subjects should age more than 18 years old, but we found the results showed in table 1, the minimum age was less than 18 years old both in Myanmar group and Laos PDR group.
  4. The English needs further editing.

Reviewer 3 Report

The topic of the paper is important and particularly timely.  The overall research is well designed, and presentation is appropriate.  Addressing a few points could improve the paper.

  1. The paper lacks a formal statement of specific aims (line 79)
  2. The authors note that participants gave informed consent (line 83), but do not indicate that the research protocol was approved by an ethics board.  This information should be added.
  3. The procedures for translation are not described in detail (lines 89-101).
  4. The exclusion of persons with specific diseases (line 127) should be justified.

Round 2

Reviewer 2 Report

The authors have already addressed all of my comments, and no additional comments.